# Skin Manifestations in COVID-19: Prevalence and Relationship with Disease Severity

**DOI:** 10.3390/jcm9103261

**Published:** 2020-10-12

**Authors:** Priscila Giavedoni, Sebastián Podlipnik, Juan M. Pericàs, Irene Fuertes de Vega, Adriana García-Herrera, Llúcia Alós, Cristina Carrera, Cristina Andreu-Febrer, Judit Sanz-Beltran, Constanza Riquelme-Mc Loughlin, Josep Riera-Monroig, Andrea Combalia, Xavier Bosch-Amate, Daniel Morgado-Carrasco, Ramon Pigem, Agustí Toll-Abelló, Ignasi Martí-Martí, Daniel Rizo-Potau, Laura Serra-García, Francesc Alamon-Reig, Pilar Iranzo, Alex Almuedo-Riera, Jose Muñoz, Susana Puig, José M. Mascaró

**Affiliations:** 1Dermatology Department, Hospital Clinic of Barcelona, University of Barcelona, Barcelona 08036, Spain; pgiavedo@clinic.cat (P.G.); podlipnik@clinic.cat (S.P.); ifuertes@clinic.cat (I.F.d.V.); ccarrera@clinic.cat (C.C.); cristina.af20@gmail.com (C.A.-F.); juditsanzb@gmail.com (J.S.-B.); MCRIQUELME@clinic.cat (C.R.-M.L.); jrieramonroig@gmail.com (J.R.-M.); andreacombalia@gmail.com (A.C.); xavi_bosch92@hotmail.com (X.B.-A.); morgadodaniel8@gmail.com (D.M.-C.); ramon.pigem@gmail.com (R.P.); atoll@clinic.cat (A.T.-A.); ignasi.marti.marti@gmail.com (I.M.-M.); danielrizopotau@msn.com (D.R.-P.); lserra94@gmail.com (L.S.-G.); francescalamonreig@gmail.com (F.A.-R.); piranzo@clinic.cat (P.I.); 2Infectious Diseases Department, Hospital Clinic of Barcelona, University of Barcelona, Barcelona 08036, Spain; pericasjm@gmail.com; 3Pathology Department, Hospital Clinic of Barcelona, University of Barcelona, Barcelona 08036, Spain; apgarcia@clinic.cat (A.G.-H.); lalos@clinic.cat (L.A.); 4International Health Department ISGlobal, Hospital Clinic, University of Barcelona; Barcelona 08036, Spain; aalmuedo@gmail.com (A.A.-R.); JMUNOZG@clinic.cat (J.M.); 5CIBER de Enfermedades Raras, Instituto de Salud Carlos III, Barcelona 08036, Spain; spuig@clinic.cat

**Keywords:** COVID-19, skin lesions, chilblain, histopathology, coronavirus, pandemic

## Abstract

Background: Data on the clinical patterns and histopathology of SARS-CoV-2 related skin lesions, as well as on their relationship with the severity of COVID-19 are limited. Methods and Materials: Retrospective analysis of a prospectively collected cohort of patients with SARS-CoV-2 infection in a teaching hospital in Barcelona, Spain, from 1 April to 1 May 2020. Clinical, microbiological and therapeutic characteristics, clinicopathological patterns of skin lesions, and direct immunofluorescence and immunohistochemical findings in skin biopsies were analyzed. Results: Fifty-eight out of the 2761 patients (2.1%) either consulting to the emergency room or admitted to the hospital for COVID-19 suspicion during the study period presented COVID-19 related skin lesions. Cutaneous lesions could be categorized into six patterns represented by the acronym “GROUCH”: Generalized maculo-papular (20.7%), Grover’s disease and other papulo-vesicular eruptions (13.8%), livedo Reticularis (6.9%), Other eruptions (22.4%), Urticarial (6.9%), and CHilblain-like (29.3%). Skin biopsies were performed in 72.4%, including direct immunofluorescence in 71.4% and immunohistochemistry in 28.6%. Patients with chilblain-like lesions exhibited a characteristic histology and were significantly younger and presented lower rates of systemic symptoms, radiological lung infiltrates and analytical abnormalities, and hospital and ICU admission compared to the rest of patients. Conclusion: Cutaneous lesions in patients with COVID-19 appear to be relatively rare and varied. Patients with chilblain-like lesions have a characteristic clinicopathological pattern and a less severe presentation of COVID-19.

## 1. Introduction

Coronavirus disease 2019 (COVID-19), caused by severe acute respiratory syndrome coronavirus 2 (SARS-CoV-2), has rapidly spread to acquire pandemic proportions since the first outbreak was declared in Wuhan, the capital of Hubei Province, China, in December 2019 [1].

Although the most common manifestations of COVID-19 are fever and respiratory symptoms such as cough and shortness of breath, other manifestations are also relevant, and subacute manifestations such as organizing pneumonia and decreased pulmonary function, or drug interactions and side effects are increasingly gaining attention as the knowledge on COVID-19 pathophysiology and natural history accumulates [2,3,4,5,6,7,8,9].

Recently, skin lesions have been described as potential manifestations of COVID-19. The skin changes reported to date include erythematous rash, urticaria, livedo reticularis, vesicular lesions, and chilblain-like lesions [10,11,12,13,14,15]. However, there are insufficient data on the prevalence and histopathology of skin lesions associated with COVID-19, as well as the associated clinical, analytical, and radiological findings.

This study was aimed to characterize the prevalence, clinical features, and histopathology findings of COVID-19-associated skin manifestations and their relationship with other COVID-19 clinical-epidemiological features.

## 2. Methods

### 2.1. Design

Single-center prospective cohort study carried out from 1 April to 1 May 2020. This study was performed following the STROBE guidelines [16].

### 2.2. Setting

Hospital Clínic de Barcelona is an 800-bed tertiary university hospital providing care to 600,000 people in the metropolitan area of Barcelona, Spain.

### 2.3. Patients

Consecutive patients with diagnosis of COVID-19 presenting new-onset of skin lesions. All consultations to the Dermatology Department originated from the Emergency Department, hospital wards, or intensive care units. The performance of skin biopsies was assessed in all patients and performed reaching a consensual agreement between the patient and the treating physician. Histologic studies requested included hematoxylin and eosin (HE) stain, direct immunofluorescence (DIF), and immunohistochemistry (IHC). DIF was performed on cryostat sections using FITC-conjugated antibodies to IgG, IgM, IgA, C3 and fibrinogen. In addition, we tested complement C9 expression by immunohistochemistry on paraffin sections of formalin-fixed tissue. Assays for detection of SARS-CoV-2 in skin samples were not done.

Real-time polymerase chain reaction (rt-PCR) from nasopharyngeal swabs was carried out amplifying the betacoronavirus *E* gene and the specific SARS-CoV-2 *RdRp* gene (Roche^®^; sensitivity 90% and specificity 100%). A serology kit was set up in our immunology laboratory for the detection of IgA, IgM or IgG SARS-CoV-2-specific antibodies (sensitivity 97% for IgA and IgG, 75% for IgM, and specificity of 100% for IgG and IgM and 98% for IgA). Diagnosis of COVID-19 included confirmed cases with positive microbiological tests and highly suspected cases, which were managed according to our institutional protocols as being infected (isolation measures ± antiviral therapies) based on highly suggestive epidemiological, clinical, radiological, and analytical features. See in Appendix A.

### 2.4. Ethics

The Hospital Clínic Ethical Board approved this study (HCB/2020/0581) and waived the requirement for informed consent due to the ongoing situation of infectious disease emergency.

### 2.5. Statistical Analysis

A descriptive analysis of the characteristics of patients presenting skin lesions associated with COVID-19 was carried out, as well as a univariate analysis of selected categories comparing patients with chilblain-like lesions and patients with other type of lesions. Categorical variables were summarized as percentages and compared through the Pearson Chi-square test or Fisher exact test when appropriate. Continuous variables were summarized as median and interquartile ranges (IQR) and compared using ANOVA test or *T* test. *p* values < 0.05 were considered statistically significant. Computing environment R (v 4.0.0, Foundation for Statistical Computing, Vienna, Austria) was used.

## 3. Results

During the study period, a total of 2761 patients either consulted to the Emergency Department (593) or were admitted to the hospital (2168) with a COVID-19 suspicion (Figure 1). In the same period, the Dermatology Department received 110 consultations, of which 70 had initial suspicion of COVID-19 related skin lesions. The final sample size consisted of 58 patients (2.1% of overall patients and 52.7% of dermatologic consultations) with diagnosis of COVID-19.

Table 1 shows a summary of the patients’ characteristics, both general and according to the clinicopathological patterns (see Appendix A (Appendix A) for an individually based account of patients’ features, including histologic findings). Median age was 54.8 years (IQR, 38.7-69) and 46.6% were female. The chronological onset of cutaneous symptoms with respect to other type of symptoms and treatments are shown in Figure 2.

Skin lesions were categorized into the following patterns: 17 chilblain-like (29%), 12 generalized maculo-papular eruptions (21%), 8 papulo-vesicular eruptions (14%), 4 livedo reticularis (7%), 4 urticarial eruptions (7%), and 13 other types of eruptions (22%). See Figure 3 and Appendix A (Appendix A) in Selected Case Studies in Appendix A.

Skin biopsies were performed in 42 (72%) patients and included DIF in 30 (52%), and IHC in 35 (60%).

Table 2 shows a comparison of the characteristics of patients with chilblain-like versus the other types of cutaneous lesions. Patients with chilblain-like lesions were significantly younger; had significantly lower percentages of microbiological confirmation, respiratory and systemic symptoms, and radiological and analytical abnormalities; were less frequently treated; and had lower hospital and ICU admission rates. In all cases except one, skin lesions appeared after the classic symptoms of COVID-19 (Figure 2). No specific association with autoantibodies was found. Histologic examination could be done in 12 of the 17 (70.6%) patients with chilblain-like lesions. In addition, 11 (64.7%) of them underwent DIF: in 7 of them (63.6%), deposition of IgM or IgG, C3, and fibrinogen on dermal blood vessels suggested a vasculopathic pattern. This was confirmed by the presence of vascular C9 deposits by IHC observed in most of these patients (90.9%, 10 of the 11 studied cases).

PCR tests on chilblains patients were performed in all cases when the “other symptoms associated with COVID-19” had already been resolved. In patients who presented without other symptoms associated with COVID-19, PCR was performed at least one week after the chilblains’ appearance. In cases where serology was performed, it was done at the same time as PCR.

Twelve patients presented with generalized maculo-papular eruptions (Appendix A), and histology showed spongiotic dermatitis, interface dermatitis, or mixed patterns (interface plus spongiotic dermatitis). In 4 patients with urticarial eruptions, histology was compatible with urticaria (Appendix A). These 2 clinicopathological groups of patients were frequently taking many medications, and therefore, the differential diagnosis between skin lesions related to COVID-19 infection or a drug eruption can very difficult, if not impossible, to do with the exception of the few patients that were not taking any medication.

Eight patients presented with papulo-vesicular eruptions on the trunk. In three of them, histology showed intraepidermal vesicles with suprabasal acantholysis with the presence of dyskeratotic keratinocytes, typical of Grover’s disease (Appendix A). In addition, there were 2 patients with chickenpox, 1 patient with herpes zoster, and 2 patients with *Pityrosporum* folliculitis. The time from the beginning of respiratory symptoms to the appearance of infectious lesions was variable. Four and 11 days for the two patients with folliculitis, 14 days for the patient with shingles, and 14 days for one of the patient’s chickenpox. The other patient with chickenpox started with skin lesions at the same time as with respiratory symptoms.

Among the 58 patients included, 39 presented cough and dyspnea. Regarding these respiratory symptoms, which were the most frequently found about the COVID-19 infection, skin lesions’ time of appearance was variable. Acral lesions (*n* = 6) appeared in a median time of 10.5 days (range: 4–26.8 days) after respiratory symptoms. In the rash maculopapular group (*n* = 8), these lesions appeared in a median of 18 days (range 13.2–24.5 days). Papulo-vesicular lesions (*n* = 7) were observed in a median of 11 days (range 7–18 days). The group of urticarial lesions (*n* = 3) appeared in a median of 28 days (range 28–35.5 days). Lesions of livedo-reticularis (*n* = 4) appeared in a median of 23 days (range 13.8–29.5 days), and in the group of other lesions, they appeared in a median of 19 days (range 9.5–33.5 days).

## 4. Discussion

Our findings suggest that skin lesions are a relatively uncommon manifestation of COVID-19 and emphasize the importance of histopathology in the characterization of skin lesions during the COVID-19 pandemic. The availability of skin biopsies, together with the specific traits of chilblain-like patterns in COVID-19, appears to be the cornerstone of COVID-19-associated skin lesions, allowing for their characterization and particularly their differentiation from other entities. Moreover, differentiating chilblain-like lesions from idiopathic and lupus-associated chilblain lesions, and other types of lesions such as generalized maculopapular or urticarial eruptions from drug reactions, is essential, given the frequency and characteristics of the former and the difficulties posed by the latter due to the large number of drugs usually concomitantly received by COVID-19 patients.

Inspired by Galván-Casas et al. proposed classification [17], we divided the cutaneous lesions into the following six groups: (1) Chilblain-like lesions, (2) Generalized maculopapular eruptions, (3) Grover’s disease and other papulo-vesicular eruptions, (4) Livedo reticularis, (5) Urticarial eruptions, (6) Other eruptions: lesions that did not meet any of the previous criteria. These skin manifestations are represented by the acronym “GROUCH”: Generalized maculo-papular; Grover’s disease and other papulo-vesicular; livedo Reticularis; Other eruptions; Urticarial; and CHilblain-like.

This study provides a comprehensive picture of skin lesions of COVID-19 supported by histopathology studies, including DIF and IHC. Compared to other studies published to date, ours has four main strengths. First, our study provides histopathology data in the majority of cases. To date, just short series of cases have provided data on biopsy findings in all patients, whereas others only provided this information in a small proportion of patients [15,17,18]. Second, we provide an accurate description of the prevalence of skin lesions in COVID-19 patients. Third, all the cases included in our study had microbiologically-confirmed or highly-suspected COVID-19, whereas the vast majority of prior reports, with some exceptions such as that of Marzano and colleagues’ study [18], included a notable proportion of patients that might not have been infected by SARS-CoV-2. Fourth, the detailed information on clinical manifestations other than cutaneous, radiological findings, analytical parameters, and particularly comprehensive histopathology data including immunofluorescence and IHC allowed us to better characterize the chilblain-like pattern as having largely specific features.

A large proportion of patients in our study did not fall in any of the 5 categories described by Galván-Casas and colleagues [17] and nonetheless presented interesting features that might be related to specific pathophysiological pathways triggered by SARS-CoV-2. For instance, four of these patients presented with acro-ischemia, and in two of them, these changes seemed to be related to vasoactive drug use; clear predisposing factors were not found in the other two cases. In a series of seven cases with acral necrosis, alterations in coagulation were observed, as well as four specific criteria of disseminated intravascular coagulation [19]. Zhang and colleagues described acro-ischemia in the context of antiphospholipid syndrome triggered by COVID-19 [20]. We found other types of vasculopathy in our series, as one case of retiform purpura with necrotic areas and three cases of livedo reticularis. Cases of transient livedo reticularis have been described in patients with COVID-19, but histologic studies were not performed [21]. Livedo reticularis can be idiopathic or associated with neoplasms, autoimmune or infectious diseases, among others, and it is also frequently observed in states of hypercoagulability [22]. One of the patients in our series presented concurrent pulmonary embolism and cutaneous intravascular thrombi, whereby DIF showed the deposition of IgM, C3, and fibrinogen within superficial-to-deep dermal blood vessel walls. In addition, C9 deposition was also demonstrated on the vessel walls by IHC [23].

Chilblain-like lesions related to COVID-19 infection have been mostly described in children and young adults [12,15,24,25]. In line with previous reports, patients with chilblain-like lesions in our series were younger, rarely presented systemic symptoms, and presented significantly fewer blood tests and radiological abnormalities compared with patients presenting other type of skin lesions. In addition, we found that these patients rarely required hospital admission and only exceptionally ICU admission. In our series, 41% of the patients with chilblain-like lesions had a confirmed diagnosis of COVID-19 whereas most cases reported elsewhere did not report this information. It is noteworthy that many of the chilblain’s patients did not have positive rt-PCR or serology. rt-PCR can be negative because chilblain occurs several days after systemic symptoms (when rt-PCR could have been negative). In other cases, chilblains appear in isolation without other symptoms, and these patients may not already have the virus present in their pharynx. Serology is negative because perhaps these clinical manifestations are mediated by cellular immunity and do not produce circulating antibodies against COVID-19. All these are hypotheses that have not been demonstrated. We believe that more studies are needed to understand the physiopathology of cashew nuts related to COVID-19.

In one study, the dermatological characteristics of 132 acral lesions in patients with suspected COVID-19 were described; skin biopsies, however, were not performed, and only 18.1% of patients had a definite COVID-19 diagnosis [14]. It has been hypothesized that these lesions begin as erythematous-violaceous macules-papules that evolve to produce subsequent blisters or digital swelling; we did not, however, observe this evolutionary pattern. The performance of histologic studies in patients with chilblain-like lesions in our series showed characteristic features in COVID-19-related chilblain-like lesions. In the HE examination, patients with chilblain-like lesions showed intense perivascular (lymphocytic vasculitis) and peri-eccrine lymphocytic infiltrations that, in many cases, also affected the subcutaneous tissue, as previously described [24,25]. In addition, prominent lichenoid dermatitis and abundant dermal and hypodermal mucin deposition could be seen. DIF showed immunoglobulin M (or G), complement, and fibrinogen deposits in the dermal blood vessels; and vascular C9 deposits were observed by IHC in most patients. In addition, CD3/CD4 positive lymphocytes with small aggregates of CD123 positive cells within inflammatory infiltrates could be observed in some of the cases. Many of these histological findings can be found both in idiopathic chilblain or in chilblain lupus, but the presence of abundant mucin deposition is more suggestive of chilblain lupus [25]. However, the climatological conditions (spring) were very unusual for chilblain, and none of these patients either had other clinical manifestations compatible with lupus nor positive anti-Ro antibodies, therefore suggesting that these lesions were indeed induced by SARS-CoV-2. Remarkably, in the histologic study on necropsies of patients who died due to COVID-19, Varga and colleagues found viral particles inside endothelial cells along with an accumulation of inflammatory cells [26]. In our study, we found that most chilblain-like lesions had a vasculopathic pattern, with DIF and IHC findings suggestive of complement pathway activation. We can hypothesize that in the setting of COVID-19 these changes may be secondary to the arrival of viral particles to the distal circulation. [27,28] Another interesting observation is that patients with monogenic type I interferonopathies (familial chilblain lupus, Aicardi-Goutières syndrome) that lead to type I interferon overproduction develop chilblain-like lesions [29]. Therefore, an exciting hypothesis to explain this type of lesions in patients with COVID-19 would be that specific immunologic repertoires present prior to SARS-CoV-2 could lead to different clinicopathological presentations. Patients who are able to develop an intense interferon response to the virus will develop mild systemic infection and chilblain-like lesions, while patients who are unable to produce interferon will develop a severe infection [27,30].

Finally, changes suggesting viral infection have been found in previous reports of papulo-vesicular lesions in COVID-19 patients [14,15,18]. In our series, in three of these patients, histology was suggestive of Grover’s disease, while the others had chickenpox, herpes zoster or *Pityrosporum* folliculitis. We believe that a proportion of COVID-19-associated skin lesions presenting a papulo-vesicular pattern correspond to the “pseudo-herpetic” variant of Grover’s disease [31,32,33]. This is particularly relevant in the differential diagnosis with other viral-induced vesicular lesions.

The main limitation of the study was a short inclusion period, therefore preventing us to provide a complete epidemiology description of COVID-19-associated lesions. Moreover, although this study provides detailed histology data in a larger proportion of patients with COVID-19 and cutaneous lesions, skin biopsy was not performed in all patients. Furthermore, a potential selection bias should be considered, since this is a single-center study carried out in a tertiary reference hospital providing care to adults but not patients in pediatric ages. Patients with skin lesions potentially related to COVID-19 with milder clinical presentations might have not been detected. Nevertheless, most patient with symptoms suggesting COVID-19 were seen at the hospital rather than the primary care during the peak of the pandemic in Barcelona, and almost all private dermatology practice was discontinued during this period as well. In addition, chilblain-like lesions associated with COVID-19 are more frequent in children and young adults; hence, the prevalence of chilblain-like lesions in our series might be lower than in the community. Finally, isolation and contact prevention measures in the hospital’s routine practices during the pandemic caused difficulties with complete skin exams. Consequently, some asymptomatic or pauci-symptomatic skin lesions might have passed unnoticed.

In conclusion, our data showed that skin lesions affect around 2% of patients with COVID-19 and can present with various patterns that can be summarized by the acronym GROUCH. Notably, patients with chilblain-like lesions have distinctive clinical and histological features and have less severe manifestations of the disease. The different clinico-pathologic patterns observed in the present study may be due to specific immunologic repertoires. Further studies are required to better define the histopathology traits, including the presence of viral particles and genetic material, as well as the immunological blueprint of COVID-19-associated skin lesions.

## Figures and Tables

**Figure 1 jcm-09-03261-f001:**
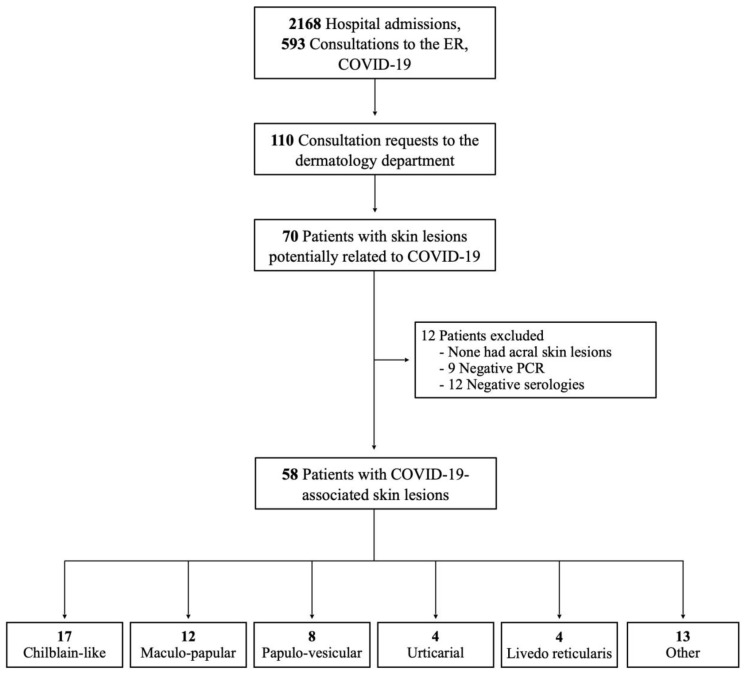
Patient flow chart according to STROBE standards.

**Figure 2 jcm-09-03261-f002:**
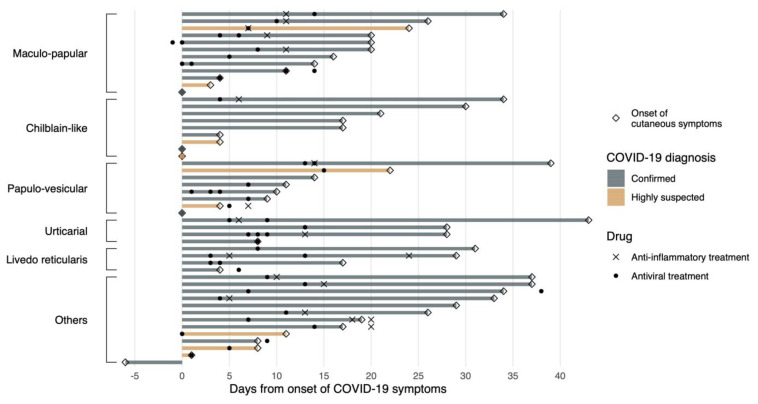
Swimmers plot showing the relationship between the onset of cutaneous manifestations with respect to other COVID-19 related symptoms. The figure also shows the initiation of treatment according to confirmed or highly suspected cases. When more than one treatment appears in a single patient, it indicates the initiation of a different type of drug (either antiviral or anti-inflammatory) along time.

**Figure 3 jcm-09-03261-f003:**
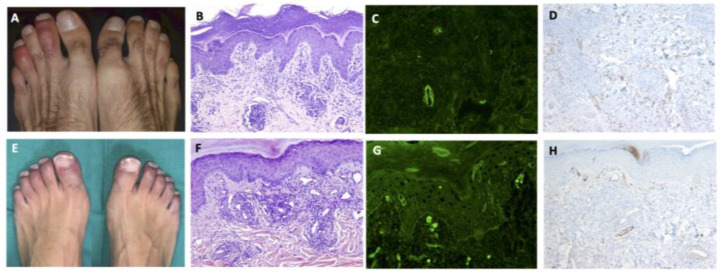
Clinical, histologic, and immunopathologic features of two cases with chilblain-like lesions. (**A**) Patient 2 Appendix A. Painful chilblain-like lesions on toes; (**B**) Lichenoid dermatitis with perivascular and periadnexal lymphocytic infiltration on superficial and deep dermis. (Hematoxylin and eosin stain, original magnification ×100); (**C**) Granular IgM deposition in dermal vessels (Direct immunofluorescence, original magnification ×200); (**D**) C9 reactivity in dermal vessels (Immunohistochemistry, original magnification ×100); (**E**) Patient 8 Appendix A. Chilblain-like violaceus lesions on toes; (**F**) Perivascular and perianexial lymphocytic infiltration on superficial and mid dermis. (Hematoxylin and eosin stain, original magnification ×100); (**G**) Granular C3 deposition in superficial dermal vessels (Direct immunofluorescence, original magnification ×200); (**H**) C9 reactivity in dermal vessels (Immunohistochemistry, original magnification ×100).

**Table 1 jcm-09-03261-t001:** Basal characteristics of the 58 patients included in the study according to the dermatological skin patterns.

Patient characteristics	Chilblain-Like (*n* = 17)	Generalized Maculo-Papular Eruption (*n* = 12)	Grover´s Disease and Other Papulo-Vesicular Eruptions (*n* = 8)	Urticarial Eruption (*n* = 4)	Livedo Reticularis (*n* = 4)	Other * (*n* = 13)	Total (*n* = 58)
Female sex, *n* (%)	7 (41%)	6 (50%)	4 (50%)	3 (75%)	1 (25%)	6 (46%)	27 (47%)
Median age, years (IQR)	29 (24.8, 47.4)	61 (51, 71.7)	48 (37.6, 61.8)	67 (51.3, 78.1)	72 (64.7, 75.3)	67 (59.4, 71.7)	55 (38.7, 69.3)
Immunosuppression, *n* (%)	1 (6%)	1 (8%)	0	0	0	2 (15%)	4 (7%)
Hypertension, *n* (%)	0	2 (17%)	3 (37%)	1 (25%)	2 (50%)	4 (31%)	12(21%)
COVID-19 diagnosis, *n* (%)							
Confirmed	7 (41%)	10 (83%)	6 (75%)	4 (100%)	4 (100%)	10 (77%)	41 (71%)
rt-PCR positive	3 (43%)	9 (90%)	4 (67%)	3 (75%)	3 (75%)	7 (64%)	29 (71%)
Serology positive	4 (57%)	2 (20%)	2 (100%)	1 (25%)	1 (25%)	3 (100%)	13 (32%)
Highly suspected	10 (59%)	2 (17%)	2 (25%)	0	0	3 (23%)	17 (29%)
COVID-19 symptoms other than cutaneous, *n* (%)	9 (53%)	12 (100%)	8 (100%)	4 (100%)	4 (100%)	13 (100%)	50 (86%)
Chest X-ray performed, *n* (%)	5	11	8	4	4	13	44
Normal	3 (60%)	1 (9%)	1 (12%)	1 (25%)	0	1 (8%)	7 (16%)
Unilateral interstitial infiltrates	0	0	0	0	0	2 (16.7%)	2 (4.5%)
Bilateral interstitial infiltrates	2 (40%)	10 (91%)	6 (75%)	3 (75%)	4 (100%)	8 (67%)	33 (75%)
Other	0	0	1 (12%)	0	0	1 (8%)	2 (4%)
Blood test with altered values suggesting COVID-19, *n* (%)	3 (19%)	11 (100%)	7 (87%)	2 (100%)	3 (100%)	13 (100%)	39 (74%)
Skin biopsy performed, *n* (%)	12 (71%)	11 (92%)	6 (75%)	4 (100%)	4 (100%)	5 (38%)	42 (72%)
DIF performed, *n* (%)	11 (65%)	3 (25%)	4 (50%)	3 (75%)	4 (100%)	5 (38%)	30 (52%)
“Antiviral” treatment, *n* (%)	1 (6%)	10 (83%)	6 (75%)	4 (100%)	4 (100%)	11 (85%)	36 (62%)
Anti-inflammatory treatment, *n* (%)	1 (6%)	10 (83%)	6 (75%)	4 (100%)	4 (100%)	11 (85%)	36 (62%)
Hospitalization required	2 (11.8%)	10 (83.3%)	6 (75%)	4 (100%)	4 (100%)	12 (92.3%)	38 (65.5%)
ICU required	1 (6%)	4 (33%)	3 (37%)	3 (75%)	2 (50%)	6 (46%)	19 (33%)
In-hospital mortality	0	0	6 (75%)	0	0	2 (5%)	2 (3%)

rt-PCR: Real-time polymerase chain reaction; DIF: Direct immunofluorescence; ICU: intensive care unit. * This group includes: pressure-induced ischemic necrosis in prolonged coma patient (*n* = 2), hematoma (*n* = 1), lichen planus (*n* = 1), contact dermatitis (*n* = 2), psoriasis (*n* = 1), generalized fixed drug eruption (*n* = 1), benign familial pemphigus (*n* = 1), chronic graft-versus-host disease (*n* = 1), stasis dermatitis (*n* = 1), dermatophytosis (*n* = 1), eruptive cherry angiomas (*n* = 1).

**Table 2 jcm-09-03261-t002:** Comparison of the characteristics of patients presenting with Chilblain-like. lesions and other types of skin lesions.

Patient characteristics	Chilblain-Like (*n* = 17)	Others (*n* = 41)	*p* Value
Female sex, *n* (%)	7 (41%)	20 (49%)	0.773
Median age, years (IQR)	29 (24, 47)	63 (50, 74)	<0.001
Immunosuppression, *n* (%)	1 (6%)	3 (7%)	1.000
Hypertension, *n* (%)	0	12 (29%)	0.012
ACE-i and ARBs use	0	6 (15%)	0.166
COVID-19 diagnosis, *n* (%)			0.003
Confirmed	7 (41.2%)	34 (82.9%)	
Highly suspected	10 (58.8%)	7 (17.1%)	
COVID-19 symptoms other than cutaneous, *n* (%)			
Fever	4 (23%)	35 (85%)	<0.001
Cough	6 (35%)	26 (63%)	0.050
Dyspnea	1 (6%)	23 (56%)	<0.001
Asthenia	1 (6%)	15 (37%)	0.023
Myalgias	1 (6%)	14 (34%)	0.045
Diarrhea	1 (6%)	12 (29%)	0.082
Ageusia	2 (12%)	8 (19%)	0.707
Anosmia	2 (12%)	5 (12%)	1.000
Headache	0	4 (10%)	0.310
Vomit	0	3 (7%)	0.548
Chest X-ray performed, *n* (%)	5 (29%)	39 (95%)	<0.0010.060
Normal	3 (60%)	4 (10%)	
Unilateral interstitial infiltrates	0	2 (5%)	
Bilateral interstitial infiltrates	2 (40%)	31 (79%)	
Other	0	2 (5%)	
**COVID-19 related parameters**			
C-Reactive protein, median mg/dL (IQR)	0.4 (0.4, 0.4)	5.9 (3.0, 14.9)	0.011
Lymphocytes, median cells ^10^6^/L (IQR)	1800 (1200, 2000)	800 (600, 1200)	<0.001
Ferritin, median ng/mL (IQR)	87.5 (40, 197.2)	615 (232.5, 905.2)	0.013
LDH, median U/L (IQR)	177 (164.2, 190.8)	299 (243.0, 415.0)	0.120
D-dimer, median ng/mL (IQR)	200 (200, 300)	800 (500, 2000)	0.297
ESR, median mm/h (IQR)	5 (4, 7)	38.0 (29, 61.8)	<0.001
Autoimmunity test performed, *n* (%)			
Anti-nuclear antibodies	15 (88.2)	22 (53.6)	
Positive	3 (20%)	1 (4%)	0.283
Anti-Ro antibodies	12 (70%)	20 (49%)	
Positive	0	0	NC
Antibeta-2-glycoprotein antibodies, IgM	15 (88.23)	24 (58.53)	
Positive	0	0	NC
Antibeta-2-glycoprotein antibodies, IgG	15 (88.2)	24 (58.5)	
Positive	1 (7%)	2 (8%)	1.000
Anticardiolipin antibodies, IgM	15 (88.2)	24 (58.5)	
Positive	0	0	NC
Anticardiolipin antibodies, IgG	15 (88.2)	24 (58.5)	
Positive	2 (13%)	2 (8%)	0.631
Skin biopsy performed, *n* (%)	12 (77%)	30 (73%)	1.000
Histological pattern according to HE stains			0.003
Chilblain-like pattern	10 (83%)	0	
Spongiotic dermatitis	0	8 (27%)	
Interface + spongiotic dermatitis	0	6 (20%)	
Grover´s disease pattern	0	3 (10%)	
Urticariform	0	2 (7%)	
Interface dermatitis	0	2 (7%)	
Pityrosporum folliculitis	0	2 (7%)	
Subcorneal pustulosis	0	2 (%)	
Psoriasiform	1 (8%)	0	
Thrombotic microangiopathy	1 (8%)	1 (3%)	
Other	0	4 (13%)	
Mucin deposits	9 (75%)	1 (3%)	<0.001
Epidermal atrophy	3 (25%)	2 (7%)	0.131
DIF performed, *n* (%)	11 (65%)	19 (46%)	0.024
Lichenoid	0	2 (10.5%)	
Lichenoid + vasculopathic	3 (27%)	0	
Negative	4 (36%)	14 (74%)	
Vasculopathic	4 (36%)	3 (15.8%)	<0.001
IHQ Anti-C9 deposits in dermal vessels performed, *n* (%)	11 (64.7)	22 (53.5)	
Positive	10 (91%)	5 (23%)	<0.001
Treatment			
Hydroxychloroquine	1 (6%)	35 (85%)	<0.001
Azithromycin	0	34 (83%)	<0.001
Lopinavir/ritonavir	1 (6%)	29 (71%)	<0.001
Systemic corticosteroids	1 (6%)	24 (58%)	<0.001
Tocilizumab	1 (6%)	14 (34%)	0.045
Remdesivir	0	5 (12%)	0.308
Anakinra	0	5 (12%)	0.308
Siltuximab	0	2 (5%)	1.000
Hospitalization required	2 (12%)	36 (88%)	<0.001
ICU required	1 (6%)	18 (44%)	0.005
Mortality	0	2 (5%)	1.000

ACEi: angiotensin-converting enzyme inhibitors; ARBs: Angiotensin II receptor antagonists; ESR: erythrocyte sedimentation rate; H/E: hematoxylin-eosin; DIF: Direct immunofluorescence; IHQ: immunohistochemistry; ICU: intensive care unit; NC: Not calculable.

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
