# Peer review of "Skin Manifestations in COVID-19: Prevalence and Relationship with Disease Severity"

_jcm, 2020, doi:10.3390/jcm9103261_

Round 1

Reviewer 1 Report

This is a comprehensive study of the skin manifestations of COVID-19 in a relatively large number of patients. The systematic study done over a one month study provides insight into the frequency of skin manifestations in COVID-19. 

Minor changes:

line 171. Inspired by (not on).

Line 179. Change to “First, our study provides…”

Line 239. Change to: “specific immunologic repertoires present prior to…”

Line 256. Patient with symptoms suggesting COVID-19 were seen at the hospital…”

Line 258.  Change to “In addition, chilblain-like lesions associated with…”

Line 261. Change to “…during the pandemic caused difficulties with complete skin exams”.

Author Response

Reviewer 1:

Thank you for your revision of our article and for your remarks.

  1. Line 171. Inspired by (not on).

We have modified the text to shorten it; we have removed this expression.

  1. Line 179. Change to “First, our study provides…”

We have done so (page 11, line 341)

  1. Line 239. Change to: “specific immunologic repertoires present prior to…”

We have changed the expression according to the reviewer’ suggestion (page 16, line 678 - 679).

  1. Line 256. Patient with symptoms suggesting COVID-19 were seen at the hospital…”

Indeed, thanks (page 13, line 640 - 641).

  1. Line 258.  Change to “In addition, chilblain-like lesions associated with…”

See the suggested change in page 13, line 643.

  1. Line 261. Change to “…during the pandemic caused difficulties with complete skin exams”.

The modified sentence can be found in Page 13 and 14, line 646 - 647.

Reviewer 2 Report

This is an interesting description of 58 subjects representing all the adults with acute dermatologic conditions seen by dermatologists during a single month of the SARS-CoV-2 epidemic at a large adult hospital in Barcelona.  The series emphasizes particularly the subjects who had chilblain-like rashes.  There are a number of problems, some of which can be rectified, some of which may not be so easy to address.

The SARS-CoV-2 antibody test is described on lines 72-73, and it appears to be a test created and verified locally (?in the hospital laboratory).  The details of the evaluation of this test (sensitivity and specificity) should be fully described in the supplement.  Since almost half the patients with chilblains had unconfirmed COVID-19, the details of the testing (particularly for ALL the “highly suspected”, but unconfirmed cases) should also be presented.  That is, were they tested by PCR?  If so, when during the illness?  Likewise, did they have blood tested for antibody and, if so, what was the timing of the blood sample(s) in relation to the illness?  It is true that the large Spanish series by Galvan-Casas had many unconfirmed cases, but this manuscript comes from a single hospital with a single laboratory.  So the details of the 59% of patients with chilblains that were not confirmed infected should be presented.  Likewise for the patients who were unconfirmed in the other dermatologic categories.

When percentages are based on numbers (in the denominator) under 100, it makes no sense to express them down to the tenths (that is, one decimal place).  The percentages for these (most of those in the tables and all those on lines 99-100, 116-121), should be rounded to the nearest whole integers.

In Figure 2, all the patients are presented for all the dermatologic conditions except 8 patients with chilblains. This is not explained or discussed.  Why are these 8 patients omitted?  Were these subjects asymptomatic except for the dermatosis?

Figure 3 is much smaller than the figures in the Supplement and, in particular, is too small to see clearly the histologic findings (including those shown by immunofluorescence or immunohistologic staining).  These should be larger.

The discussion is too long.  The enormous paragraph on lines 206-242 contains much material that should be in the Results section, not in the Discussion.  That paragraph, and other parts of the Discussion, should be shortened, and the material that belongs in Results should be moved.  This will entail quite a lot of rewriting, but the paper will be very much improved.

On lines 150-153 it is stated that 5 of those with papulo-vesicular rashes had other infectious diseases (“2 chickenpox, 1 herpes zoster, and 2 Pityrosporum”).  The large supplementary table does not mention which of the 5 subjects (patients no. 33-37) had which of these other infections; it would be instructive (and important) to footnote the supplementary table to identify which subjects had viral or fungal infections.  Presumably these viral and fungal infections were identified clinically and not microbiologically.  The timing of these non-COVID infections could also be indicated in Figure 2, which would be very instructive.

The wide variation in the timing of the dermatologic manifestations in relation to the respiratory and systemic symptoms of Covid-19 is interesting and should be discussed or at least mentioned in the paper.

The “Others” category is interesting.  In this regard, the paper would be more informative if the large supplementary table identified each of these specifically, rather than simply saying “Other” in the column headed “Cutaneous presentation.”

Specific issues:

On lines 68-69 DIF and IHC are mentioned, but the antigens and/or antibodies looked for are not mentioned.  Perhaps this is obvious to a dermatologist, but this reviewer is not a dermatologist, nor will many of the readers of this paper be.  Also, it seems that SARS-CoV-2 antigens were not looked for.  This also should be specifically mentioned.

In both Table 1 and Table 2 the writing in left-hand column has been centered.  It would be clearer if the writing started at the left margin of that column, with sub-headings indented.

In Figure 2 meaning of the solid black diamonds is not defined. 

In Table 1, “Antiviral treatment” is listed.  Presumably this includes hydroxychloroquine, azithromycin, NEITHER of which is an antiviral treatment. One solution would be to use the word “antimicrobial,” with a footnote if antibiotics are not included.  Another solution would be to put “antiviral” in quotation marks.

In Supplementary Table 1, the treatments are given as abbreviations, all of which should be identified fully in footnotes.  It would also greatly help the reader if Supplementary Table 1 repeated ALL the column headings on each page.

More minor issues:

On line 50, “is” should be “are” since “data” is the plural of “datum.”  It should read: “However, there are insufficient data…”

Line 59 mentions the STROBE guidelines.  A suitable specific reference would be helpful.

On line 72, the sensitivity of the Roche PCR test is stated as 68%.  Is this a typographical error?  The Roche PCR test is the widely accepted standard way of finding this virus, and PCR, if performed during or 1-2 days before symptoms, is clearly the most sensitive way of identifying Covid-19 (i.e. sensitivity very close to 100%).  Timing, however, is important.

Line 150.  This should read, “…eruptions on the trunk”, not eruptions in the trunk.

In Table 2, in the line “Chest X-ray performed, n (%), there are 2 p-values, one slightly above and one slightly below the line.  Which one applies, and to which comparison?

On line 182, it should read, “prevalence of skin lesions in COVID-19 patients.”

On line 185, it should read, “…a notable proportion of patients that might not have been infected…”

On line 194, I suggest using a semicolon rather than a comma, and the sentence should read: “…related to vasoactive drug use; clear predisposing factors were not found in the other two cases.”

On line 208, suggest “fewer” instead of “lesser.”

On lines 214 and 216, “however” is used as a conjunction between independent clauses.  This is not the correct use of “however.”  Suggest (on line 214) putting a semicolon after “described” and the following order of words: “described; skin biopsies, however, were not performed, and only…”  Likewise, suggest: “…digital swelling; we did not, however, observe this…”

Author Response

Reviewer 2:

We appreciate your time and the thorough revision of our article.

  1. The SARS-CoV-2 antibody test is described on lines 72-73, and it appears to be a test created and verified locally (?in the hospital laboratory). The details of the evaluation of this test (sensitivity and specificity) should be fully described in the supplement.   

We have included a citation of a study carried out in our hospital that explains in detail the serology technique (García-Basteiro et al, Nat Comm 2020).

  1. Since almost half the patients with chilblains had unconfirmed COVID-19, the details of the testing (particularly for ALL the “highly suspected”, but unconfirmed cases) should also be presented. That is, were they tested by PCR?  If so, when during the illness?  Likewise, did they have blood tested for antibody and, if so, what was the timing of the blood sample(s) in relation to the illness?  It is true that the large Spanish series by Galvan-Casas had many unconfirmed cases, but this manuscript comes from a single hospital with a single laboratory.  So the details of the 59% of patients with chilblains that were not confirmed infected should be presented.  Likewise for the patients who were unconfirmed in the other dermatologic categories.

PCR tests on chilblains patients were performed when the "other symptoms associated with COVID-19" had already resolved. In patients who presented without other symptoms associated with COVID-19, PCR was performed between one and three weeks after chilblains' appearance. In cases where serology was performed, it was done at the same time as PCR. Seventeen patients presented chilblains lesions; in all cases, PCR and/or serology tests were performed. In 11 cases, both techniques were performed. We added these data in the text (Page 8, lines 252-256), where the details of the PCR tests and serologies in each group are shown. These data can also be found in the supplementary material for each patient.

Chilblains lesions (n: 17):

- PCR was performed in 13 (76.47%) cases and was positive in 3.

- Serology was performed in 15 (88.23%) patients and was positive in 4.

Ten (58.82%) patients without confirmation by PCR or serology:

-6 (60%) had findings of vasculopathy and c9 deposits in the biopsy

-2 (20%) with c9 deposits without vasculopathy,

- Only in 2 (20%) cases, no skin biopsy was performed. 

These clinical and histological findings of perniosis are highly unusual during spring months, therefore suggesting its association with COVID-19.

Papular rash lesions (n: 12): 8 had positive PCR (of 12 PCR performed), one positive serology (of the 3 completed), and one patient both positive techniques.

In one patient, both serology and PCR were negative, and in another, only PCR was performed, which was negative. These two patients had clinical and analytical tests compatible with COVID infection.19 One of them also had X-rays with bilateral infiltrates.

Papule vesicles lesions (n: 8): 6 PCR were performed, of which four were positive, and two serologies both positive. In this subgroup, 6 out of 8 patients were confirmed

Urticarial lesions (n:4): 4 had PCR were performed, of which 3 were positive. The one who had negative PCR had positive serology.

4 patients with livedo reticularis: 3 had positive PCR, and one had positive serology. Thus, all 4 cases in this group were confirmed.

13 patients with “other lesions”: PCR was done in 11 patients and was positive in 7. Serology was done in 3 patients and was positive in all 3. Thus, 10 out of the 13 patients in this group were confirmed with one of these two techniques.

In two of the 13 patients, the PCR was negative, and in these two cases, the serology was not done. In one patient in this group, no PCR or serology was performed. In these three commented cases, where neither PCR nor serology infection was confirmed, the clinical and radiographic findings were compatible with COVID-19; and two of these three patients also had analytical results consistent with COVID-19.

  1. When percentages are based on numbers (in the denominator) under 100, it makes no sense to express them down to the tenths (that is, one decimal place). The percentages for these (most of those in the tables and all those on lines 99-100, 116-121), should be rounded to the nearest whole integers.

We have modified the % in the text and the tables, as suggested by the reviewer.

  1. In Figure 2, all the patients are presented for all the dermatologic conditions except 8 patients with chilblains. This is not explained or discussed. Why are these 8 patients omitted?  Were these subjects asymptomatic except for the dermatosis?

 The eight patients with chilblains lesions not shown in Figure 2 had no other    symptoms compatible with COVID-19; this has been clarified in results (Page 7; line 219 – 220) and further added in the footnote to Figure 2

  1. Figure 3 is much smaller than the figures in the Supplement and, in particular, is too small to see clearly the histologic findings (including those shown by immunofluorescence or immunohistologic staining). These should be larger.

We increased the size of figure 3 while maintaining its resolution.

  1. The discussion is too long. The enormous paragraph on lines 206-242 contains much material that should be in the Results section, not in the Discussion.  That paragraph, and other parts of the Discussion, should be shortened, and the material that belongs in Results should be moved.  This will entail quite a lot of rewriting, but the paper will be very much improved.

Following the reviewer’ suggestions, the discussion has been largely shortened and some parts have been moved to the results’ section.

  1. On lines 150-153 it is stated that 5 of those with papulo-vesicular rashes had other infectious diseases (“2 chickenpox, 1 herpes zoster, and 2 Pityrosporum”). The large supplementary table does not mention which of the 5 subjects (patients no. 33-37) had which of these other infections; it would be instructive (and important) to footnote the supplementary table to identify which subjects had viral or fungal infections.  Presumably these viral and fungal infections were identified clinically and not microbiologically. 

The reviewer is right. In the revised version, we have indicated with a superscript number that of the five subjects (patients nº 33-37) had these other infections in the supplementary table is indicated. Moreover, the footnote now identifies which patients had viral or fungal infections. 

These viral and fungal infections were identified both clinically and microbiologically. They were determined by herpes virus PCR in herpes infections and histological findings in the case of Pityrosporum sp folliculitis.

  1. The timing of these non-COVID infections could also be indicated in Figure 2, which would be very instructive.

These data have been added in the text, (Page 7; lines 214 - 215), since the table could not be easily read.

The time from the onset of respiratory symptoms to the appearance of infectious lesions was variable: Four and 11 days for the two patients with folliculitis, 14 days for the patient with shingles, and 14 days for one of the patient's chickenpox. The other patient with chickenpox started with skin lesions at the same time as with respiratory symptoms.

  1. The wide variation in the timing of the dermatologic manifestations in relation to the respiratory and systemic symptoms of Covid-19 is interesting and should be discussed or at least mentioned in the paper.

Among the 58 patients included, 39 presented cough and dyspnea. Regarding these respiratory symptoms, which were the most frequently found about the COVID-19 infection, skin lesions' time of appearance was variable.

Acral lesions (n=6) appeared in a median time of 10.5 days (range: 4 - 26.8 days) after respiratory symptoms. In the rash maculopapular group (n=8), these lesions appeared in a median of 18 days (range 13.2 - 24.5 days). Papulo-vesicular lesions (n=7) were observed in a median of 11 days (range 7 - 18 days). The group of urticarial lesions (n=3) appeared in a median of 28 days (range 28 - 35.5 days). Lesions of livedo-reticularis (n=4) appeared in a median of 23 days (range 13.8 - 29.5 days), and in the group of other lesions, they appeared in a median of 19 days (range 9.5 - 33.5 days). We have added this detailed information in pages 6 and 7 lines 176 - 218.

  1. The “Others” category is interesting.  In this regard, the paper would be more informative if the large supplementary table identified each of these specifically, rather than simply saying “Other” in the column headed “Cutaneous presentation.”

In the category "Other" of the supplementary table, each of the diagnoses was specifically identified. This group includes: pressure-induced ischemic necrosis in prolonged coma patient (n=2), hematoma (n=1), lichen planus (n=1), contact dermatitis (n=2), psoriasis (n=1), generalized fixed drug eruption (n=1), benign familial pemphigus (n=1), chronic graft-versus-host disease (n=1), stasis dermatitis (n=1), dermatophytosis (n=1), eruptive cherry angiomas (n=1).

  1. On lines 68-69 DIF and IHC are mentioned, but the antigens and/or antibodies looked for are not mentioned. Perhaps this is obvious to a dermatologist, but this reviewer is not a dermatologist, nor will many of the readers of this paper be.  Also, it seems that SARS-CoV-2 antigens were not looked for.  This also should be specifically mentioned.

DIF was performed on cryostat sections using FITC-conjugated antibodies to IgG, IgM, IgA, C3 and fibrinogen. In addition, we tested complement C9 expression by immunohistochemistry on paraffin sections of formalin-fixed tissue. Assays for detection of SARS-CoV-2 in skin samples were not done. We have added this information in the manuscript (page 2; lines 78 – 81).

  1. In both Table 1 and Table 2 the writing in left-hand column has been centered. It would be clearer if the writing started at the left margin of that column, with sub-headings indented.

We have modified in both Table 1 and Table 2, the left column's writing starting from the left margin, with the subtitles indented.

  1. In Figure 2 meaning of the solid black diamonds is not defined. 

Black solid diamonds are overlapping images of white diamonds (beginning of skin symptoms) with black dots (beginning of treatment).

  1. In Table 1, “Antiviral treatment” is listed. Presumably this includes hydroxychloroquine, azithromycin, NEITHER of which is an antiviral treatment. One solution would be to use the word “antimicrobial,” with a footnote if antibiotics are not included.  Another solution would be to put “antiviral” in quotation marks.

In table 1, we have put "antiviral" in quotation marks.

  1. In Supplementary Table 1, the treatments are given as abbreviations, all of which should be identified fully in footnotes. It would also greatly help the reader if Supplementary Table 1 repeated ALL the column headings on each page.

In Supplementary Table 1, the treatments are presented as abbreviations; we have fully identified them in the footnotes. Also, in Supplementary Table 1, the titles of the columns on each page have been repeated.

  1. On line 50, “is” should be “are” since “data” is the plural of “datum.” It should read: “However, there are insufficient data…

On page 2; line 57, we wrote, “However, there are insufficient data…”

  1. Line 59 mentions the STROBE guidelines. A suitable specific reference would be helpful.

We have added the following reference.: Vandenbroucke JP, von Elm E, Altman DG, Gøtzsche PC, Mulrow CD, Pocock SJ, et al. Strengthening the Reporting of Observational Studies in Epidemiology (STROBE): explanation and elaboration. PLoS Med. 16 de octubre de 2007;4(10):e297.

  1. On line 72, the sensitivity of the Roche PCR test is stated as 68%. Is this a typographical error?  The Roche PCR test is the widely accepted standard way of finding this virus, and PCR, if performed during or 1-2 days before symptoms, is clearly the most sensitive way of identifying Covid-19 (i.e. sensitivity very close to 100%).  Timing, however, is important

We have corrected the PCR sensitivity value. The PCR technique has a sensitivity of 90%, when performed on patients with high clinical suspicion of having COVID-19.

  1. Line 150.  This should read, “…eruptions on the trunk”, not eruptions in the trunk.

On line 265, we have written: "...eruptions on the trunk."

  1. In Table 2, in the line “Chest X-ray performed, n (%), there are 2 p-values, one slightly above and one slightly below the line. Which one applies, and to which comparison?

Indeed. The p value above (P<.001) corresponds to the direct comparison between the two groups (chilblain-like vs. other). We have now moved the other p value (P=0.060) to the square below under the category “Chest X-ray findings), indicating the multiple comparisons among the subgroups.

  1. On line 182, it should read, “prevalence of skin lesions in COVID-19 patients.”

In Page 11; line 346 - 347, we have written "Prevalence of skin lesions in patients with COVID-19".

  1. On line 185, it should read, “…a notable proportion of patients that might not have been infected…”

In Page 11; line 350, we have written: "...a remarkable proportion of patients who might not have been infected...

  1. On line 194, I suggest using a semicolon rather than a comma, and the sentence should read: “…related to vasoactive drug use; clear predisposing factors were not found in the other two cases.”

On page 12; line 563, we have used a semicolon instead of a comma, and the phrase has been modified to: "...related to vasoactive drug use; no clear predisposing factors were found in the other two cases.

  1. On line 208, we have written “fewer” instead of “lesser.”

On page 11; line 361, we have written “fewer” instead of “lesser.”

  1. On lines 214 and 216, “however” is used as a conjunction between independent clauses.  This is not the correct use of “however.”  Suggest (on line 214) putting a semicolon after “described” and the following order of words: “described; skin biopsies, however, were not performed, and only…”  Likewise, suggest: “…digital swelling; we did not, however, observe this…”

On page 12; lines 536 – 537 and 539 - 540, we have modified the use of "however," as suggested by the reviewer.

Reviewer 3 Report

This is a prospective study aimed to characterize the prevalence, clinical features, and histopathology of patients with COVID-19-associated skin lesions. Although the authors accomplished their goal providing a comprehensive description of clinical features, the prevalence and histopathological features are less well characterized. Some important references need to be added.

The information is nicely presented and the illustrations are good. However, some important bits are missing, particularly more detailed description of histopathological features. A table could work very well to fix this lack of histopathological information. The role of histopathology in the diagnosis of these lesions is probably overestimated. Histopathology of skin lesion provides useful information to understand the pathogenesis of COVID-19 related lesions but is probably not necessary in most cases to reach a diagnosis. This is not completely clear in the text.

In the discussion section, you stated:  "And fourth, the detailed information on clinical manifestations other than cutaneous, radiological findings, analytical parameters, and particularly comprehensive histopathology data including immunofluorescence and IHC allowed us to better characterize the chilblain-like pattern as having largely specific features". What are those specific features you mentioned? Do you think you can differentiate, based on histology, chilblain-like lesions related to COVID-19 from idiopathic chilblains or lupus-chilblains? If you think so, please give more details.

I agree that the prevalence of cutaneous lesions in COVID-19 is low, even lower than demonstrated by your results. You analyze the prevalence in a selected group of patients, those who were admitted in hospital or consulted at the during the peak of the pandemic in Barcelona. The prevalence in the general population infected, but with mild symptoms, has not been analyzed.

...."we found that most chilblain-like lesions had a vasculopathic pattern, with DIF and IHC findings suggestive of complement pathway activation. We can hypothesize that in the setting of COVID-19 these changes may be secondary to the arrival of viral particles to the distal circulation"....there are papers describing the presence of viral particles in the skin of COVID-19 chilblains, please cite them (doi.org/10.1111/bjd.19415; 10.1111/bjd.19327).

"Patients who are able to develop an intense interferon response to the virus will develop mild systemic infection and chilblain-like lesions, while patients who unable to produce interferon will develop a severe infection". This hypothesis was nicely discussed in the paper by Magro et al, this important citation is missing (10.1111/bjd.19415).

Author Response

Reviewer 3:

Thank you for your review and comments.

  1. This is a prospective study aimed to characterize the prevalence, clinical features, and histopathology of patients with COVID-19-associated skin lesions. Although the authors accomplished their goal providing a comprehensive description of clinical features, the prevalence and histopathological features are less well characterized. Some important references need to be added. (doi.org/10.1111/bjd.19415; 10.1111/bjd.19327 and 10.1111/bjd.19415)

We have added the references suggested by the reviewer.

  1. The information is nicely presented and the illustrations are good. However, some important bits are missing, particularly more detailed description of histopathological features. A table could work very well to fix this lack of histopathological information.

Thank you for your comment. Unfortunately, we cannot expand the length of our manuscript. However, in its current form Table 2 details the characteristic histological findings in the group of patients with perniosis and the rest, Supplementary Table 1 provides a case-by-case description, including findings from hematoxylin-eosin, direct immunofluorescence, and immunohistochemistry studies, and Figure 3 and Supplementary figures provide clinico-histopathologic descriptions of selected cases.

  1. The role of histopathology in the diagnosis of these lesions is probably overestimated. Histopathology of skin lesion provides useful information to understand the pathogenesis of COVID-19 related lesions but is probably not necessary in most cases to reach a diagnosis. This is not completely clear in the text.

We fully agree with this comment. When we performed the work, there was little information about the histology of patients with COVID and skin lesions. We believe that, according to current knowledge, in most patients, a biopsy is not necessary to make the diagnosis. We have added this reflection in the first paragraph of the discussion.

  1. In the discussion section, you stated: "And fourth, the detailed information on clinical manifestations other than cutaneous, radiological findings, analytical parameters, and particularly comprehensive histopathology data including immunofluorescence and IHC allowed us to better characterize the chilblain-like pattern as having largely specific features". What are those specific features you mentioned? Do you think you can differentiate, based on histology, chilblain-like lesions related to COVID-19 from idiopathic chilblains or lupus-chilblains? If you think so, please give more details.

We have rephrased our assertions regarding the potentially distinctive characteristics of chillblain lesions associated to COVID-19 in the abstract, discussion and conclusions.

  1. I agree that the prevalence of cutaneous lesions in COVID-19 is low, even lower than demonstrated by your results. You analyze the prevalence in a selected group of patients, those who were admitted in hospital or consulted at the during the peak of the pandemic in Barcelona. The prevalence in the general population infected, but with mild symptoms, has not been analyzed.

The reviewer is right. We have stressed this important limitation in the limitations’ section of the discussion.

  1. ...."we found that most chilblain-like lesions had a vasculopathic pattern, with DIF and IHC findings suggestive of complement pathway activation. We can hypothesize that in the setting of COVID-19 these changes may be secondary to the arrival of viral particles to the distal circulation"....there are papers describing the presence of viral particles in the skin of COVID-19 chilblains, please cite them (doi.org/10.1111/bjd.19415; 10.1111/bjd.19327).

We have added these two references that explain the presence of the virus in the endothelium.

  1. "Patients who are able to develop an intense interferon response to the virus will develop mild systemic infection and chilblain-like lesions, while patients who unable to produce interferon will develop a severe infection". This hypothesis was nicely discussed in the paper by Magro et al, this important citation is missing (10.1111/bjd.19415).

We have also added this reference from Magro et al., which discusses this hypothesis very well.

Round 2

Reviewer 2 Report

This revision is much improved.  The paper remains very interesting.

I am still concerned about the fact that the subjects with chilblains are, as a group, the ones that have the lowest proportion with a clear diagnosis of SARS-CoV-2 infection (7 of 17).  There may be good reasons for this.  Can the authors think of a good reason for this?  If so, they should include it in the Discussion section.  If not, they can leave the paper as it is and leave it up to the reader to say to him/herself: "There seems to be no good explanation for why the subjects with chilblains in the springtime during the COVID-19 pandemic have no evidence for COVID-19; it is interesting that the authors did not comment on this."

Although the authors say they have taken out the decimal places in the percentages in the tables and the text, in fact they have not take them out of much of the text (e.g. the Abstract).  They can go through the text again and correct this easily.

Author Response

Reviewer 2:

Minor changes:

  • This revision is much improved.  The paper remains very interesting. I am still concerned about the fact that the subjects with chilblains are, as a group, the ones that have the lowest proportion with a clear diagnosis of SARS-CoV-2 infection (7 of 17).  There may be good reasons for this.  Can the authors think of a good reason for this?  If so, they should include it in the Discussion section.  If not, they can leave the paper as it is and leave it up to the reader to say to him/herself: "There seems to be no good explanation for why the subjects with chilblains in the springtime during the COVID-19 pandemic have no evidence for COVID-19; it is interesting that the authors did not comment on this."

That's a very relevant observation. It is currently unclear why patients with chilblain do not have PCR or positive serology in many cases. The PCR can be negative because the perniosis occurs several days after the systemic symptoms (when the PCR could have already been negative) or in an isolated way without other symptoms. The serology is negative because perhaps these clinical manifestations are mediated by cellular immunity and do not produce circulating antibodies against the COVID-19. All these are hypotheses that have not been demonstrated. We believe that studies are lacking to know the physiopathology of chilblain that occurred during COVID-19. We have added this reflection to the discussion.

  • Although the authors say they have taken out the decimal places in the percentages in the tables and the text, in fact they have not take them out of much of the text (e.g. the Abstract).  They can go through the text again and correct this easily.

Thanks for the observation; it was a mistake not to modify the decimals in the text. We have corrected it.